# Inhibition of miR-199b-5p reduces pathological alterations in osteoarthritis by potentially targeting *Fzd6* and *Gcnt2*

Tong Feng[1†], Qi Zhang[1,2†], Si-Hui Li[1], Yan-ling Ping[1], Mu-qiu Tian[1], Shuan-hu Zhou[3,4], Xin Wang[5], Jun-Meng Wang[1], Fan-Rang Liang[1], Shu-Guang Yu[1,6], Qiao-Feng Wu[1,6]*

[1]Acupuncture and Tuina College, Chengdu University of Traditional Chinese Medicine, Chengdu, China; [2]Chongqing Hospital of Traditional Chinese Medicine, Chongqing, China; [3]Department of Orthopedic Surgery, Brigham and Women's Hospital, Harvard Medical School, Boston, United States; [4]Harvard Stem Cell Institute, Harvard University, Cambridge, United States; [5]Departments of Neurosurgery, Brigham and Women's Hospital, Harvard Medical School, Boston, United States; [6]Key Laboratory of Acupuncture for Senile Disease (Chengdu University of TCM), Ministry of Education, Chengdu, China

*For correspondence:
wuqiaofeng@cdutcm.edu.cn

[†]These authors contributed equally to this work

Competing interest: The authors declare that no competing interests exist.

**Abstract** Osteoarthritis (OA) is a degenerative disease with a high prevalence in the elderly population, but our understanding of its mechanisms remains incomplete. Analysis of serum exosomal small RNA sequencing data from clinical patients and gene expression data from OA patient serum and cartilage obtained from the GEO database revealed a common dysregulated miRNA, miR-199b-5p. In vitro cell experiments demonstrated that miR-199b-5p inhibits chondrocyte vitality and promotes extracellular matrix degradation. Conversely, inhibition of miR-199b-5p under inflammatory conditions exhibited protective effects against damage. Local viral injection of miR-199b-5p into mice induced a decrease in pain threshold and OA-like changes. In an OA model, inhibition of miR-199b-5p alleviated the pathological progression of OA. Furthermore, bioinformatics analysis and experimental validation identified *Gcnt2* and *Fzd6* as potential target genes of *MiR-199b-5p*. Thus, these results indicated that *MiR-199b-5p/Gcnt2* and *Fzd6* axis might be a novel therapeutic target for the treatment of OA.

## eLife assessment

This **valuable** study reports that miR-199b-5p is elevated in human osteoarthritis patients. There is **solid** evidence for the finding that inhibiting miR-199b-5p alleviates symptoms in mice with knee osteoarthritis. Additionally, potential targets of miR-199b-5p are identified but whether miR-199b-5p truly functions through Fzd6 and/or Gcnt2 requires further investigation.

## Introduction

Osteoarthritis (OA) is a degenerative disease characterized by the deterioration of articular cartilage and affecting all components of the joint, including the synovium, subchondral bone, and meniscus (*Chen et al., 2017*). Knee OA (KOA) is the leading cause of disability and increased living costs among elderly patients (*Hunter and Osteoarthritis, 2019*). Currently, the treatment options for osteoarthritis are limited to symptomatic relief and joint replacement. However, the insufficient relief of symptoms, potential medication side effects, and the economic burden and complications associated with

joint replacement surgeries have prompted the need to identify new, effective, and safe treatment methods and explore unknown targets for individuals with KOA (*Ito et al., 2021*).

Exosomes are extracellular vesicles that are distributed in body fluids such as serum and play a crucial biological role by delivering molecules such as microRNAs (miRNAs) (*Xu et al., 2016*). They serve as vital carriers for intercellular communication and transfer of genetic information. miRNAs are a class of non-coding RNAs ranging from 18 to 24 nucleotides in length, which exert inhibitory effects on the expression of target genes through interactions with mRNA. Studies have demonstrated that miRNAs play a crucial role as a pathogenic factor in OA (*Ali et al., 2021*). For instance, miR-199b-5p was found to contribute to the osteogenic differentiation of bone marrow stromal cells (*Zhao et al., 2016*) another miR-140 showed cartilage-specific expression, and its expression was significantly reduced in OA cartilage (*Yamashita et al., 2012*; *Miyaki et al., 2009*). Recently, intra-articular injection of antisense oligonucleotides of miR-181a-5p produced chondroprotective effects in OA mice (*Nakamura et al., 2019*).

In this study, we initially screened miR-199b-5p as a potential key miRNA based on clinical data by detecting differentially expressed exosomal miRNAs. To elucidate the role and function of miR-199b-5p, we investigated its impact on chondrocytes and identified its molecular targets through in vitro experiments. Furthermore, we explored its role in vivo. Hence, this article not only identifies miR-199b-5p as a potential micro-target for KOA but also provides a potential strategy for future identification of new molecular drugs.

## Results
### Identification and enrichment analysis of differentially expressed miRNAs in serum exosomes

To investigate the dysregulated miRNAs in serum exosomes of KOA patients, we extracted exosomal miRNAs from serum and performed sequencing. Serum samples from 15 patients with KOA and 10 healthy subjects were collected (*Supplementary file 1*, Supplement table 2). After extraction, the serum exosomes were observed under transmission electron microscopy and nanoparticle tracking analysis, revealing a diameter ranging from approximately 70–150 nm (*Figure 1—figure supplement 1A, B*). Furthermore, the surface marker proteins CD9, CD63, and CD81 were found to be expressed on the exosomes (*Figure 1—figure supplement 1C*). Next, we performed sequencing of miRNAs in serum exosomes.

The results showed that 88 miRNAs were up-regulated and 89 miRNAs were downregulated in KOA patients compared with the control group based on fold change >1.5 and p<0.05 (*Figure 1A and B*). Afterwards, we performed bioinformatics Gene Ontology (GO) and Kyoto Encyclopedia of Genes and Genomes (KEGG) analyses on these differentially expressed miRNA. Among the top five enriched results of up-regulated differentially expressed miRNAs, the CC (Cell GO:0044297, intracellular GO:0005622, and cytoplasm GO:0005737) GO terms were notable. The MF (Translation activator activity GO:0008494, binding to ubiquitin-conjugating enzymes GO:0031624, and possessing transferase activity GO:0016740) GO terms, and BP (Cell differentiation GO:0045595 and the stabilization of mRNA through 3'-UTR mediation GO:0070935) GO terms were also significant (*Figure 1C*). The top 10 enriched KEGG pathways, PI3K-AKT signaling, Ras signaling, apoptosis, and other types of O-glycan biosynthesis are considered to be of significant importance (*Figure 1D*). The analysis of the top five down-regulated differential miRNAs reveals interesting findings. the CC (Cell GO:0005623, intracellular GO:0005622) GO terms were particularly noteworthy. The MF (RNA polymerase II transcription factor activity and sequence-specific DNA binding GO:0000981) GO terms, as well as the BP (collagen-activated signaling pathway GO:0038065) GO terms, were also deemed significant (*Figure 1E*). Moreover, the analysis of the top 10 enriched KEGG pathways revealed a strong association with the Hedgehog signaling pathway, pluripotency of stem cells, and glycosaminoglycan biosynthesis-keratan sulfate (*Figure 1F*).

It is noteworthy that in the aforementioned results, both the up-regulated and down-regulated differential miRNA enrichment analyses are involved in processes related to cartilage synthesis, including O-glycan biosynthesis and glycosaminoglycan biosynthesis-keratan sulfate (*Wu et al., 2023*; *Yang et al., 2008*).

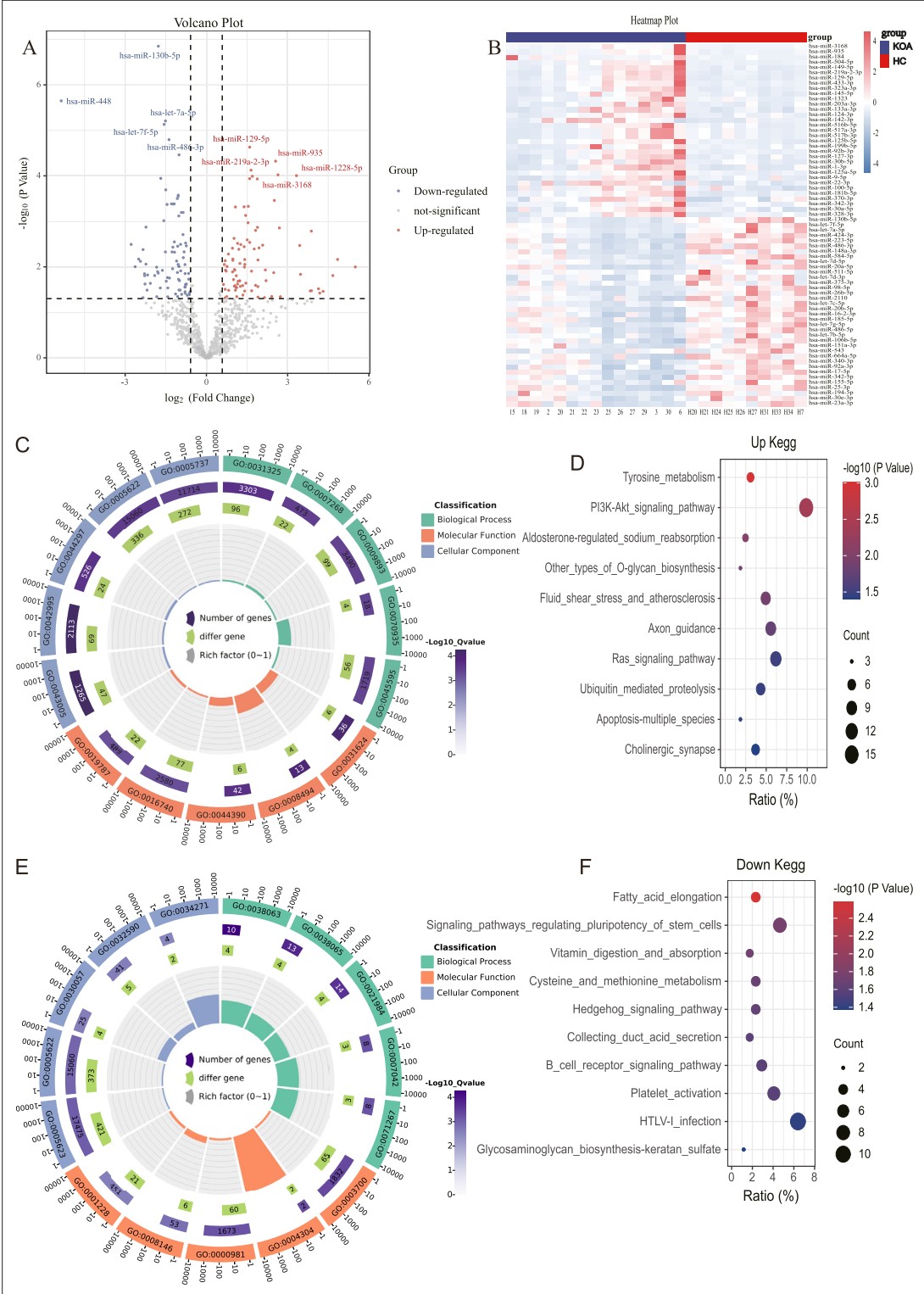

**Figure 1.** MicroRNA (miRNA) expression in knee osteoarthritis (KOA) patients. (**A, B**) Volcano plot and heatmap showing differential miRNAs between KOA and healthy groups (healthy, n=10; KOA, n=15, |log2 Fold Change|≥0.585, p<0.05). (**C**) GO enrichment analyses of upregulated genes.(GO:0031325 positive regulation of cellular metabolic process; GO:0007268 chemical synaptic transmission; GO:0009893 positive regulation of metabolic process; GO:0070935 3'-UTR-mediated mRNA stabilization; GO:0045595 regulation of cell differentiation; GO:0043005 neuron projection; GO:0042995 cell projection; GO:0044297 cell body; GO:0005622 intracellular; GO:0005737 cytoplasm; GO:0031624 ubiquitin-conjugating enzyme binding; GO:0008494 translation activator activity; GO:0044390 ubiquitin-like protein conjugating enzyme binding; GO:0016740 transferase activity; GO:0019787 ubiquitin-

*Figure 1 continued on next page*

*Figure 1 continued*

like protein transferase activity) (**D**) Kyoto Encyclopedia of Genes and Genomes (KEGG) enrichment analyses of upregulated genes. (**E**) Gene Ontology (GO) enrichment analyses of downregulated genes. (GO:0038063 collagen-activated tyrosine kinase receptor signaling pathway; GO:0038065 collagen-activated signaling pathway; GO:0021984 adenohypophysis development; GO:0007042 lysosomal lumen acidification; GO:0071267 L-methionine salvage; GO:0005623 cell; GO:0005622 intracellular; GO:0030057 desmosome; GO:0032590 dendrite membrane; GO:0034271 phosphatidylinositol 3-kinase complex, class III, type I; GO:0003700 DNA binding transcription factor activity; GO:0004304 estrone sulfotransferase activity; GO:0000981 RNA polymerase II transcription factor activity, sequence-specific DNA binding; GO:0008146 sulfotransferase activity; GO:0001228 transcriptional activator activity, RNA polymerase II transcription regulatory region sequence-specific DNA binding) (**F**) KEGG enrichment analyses of downregulated genes.

The online version of this article includes the following source data and figure supplement(s) for figure 1:

**Figure supplement 1.** Identification of exosome.

**Figure supplement 1—source data 1.** Original file of western blot in *Figure 1—figure supplement 1*.

**Figure supplement 1—source data 2.** Labelled file of western blot in *Figure 1—figure supplement 1*.

## Dysregulated serum exosomal miRNAs were found to be expressed in both serum and cartilage

Subsequently, we aimed to investigate whether these dysregulated miRNAs in serum exosomes show differential expression in other affected sites or tissues of KOA. Leveraging the KOA patient cartilage and serum sequencing data available in the GEO database, we compared our dysregulated miRNA list with the datasets. Remarkably, we identified 169 miRNAs that exhibited differential expression in the serum of KOA patients (*Figure 2A*), suggesting their involvement in the disease. Moreover, these miRNAs were also found to be expressed in KOA patient cartilage (*Figure 2B*). This robust validation reinforces the reliability of our data. Consequently, we proceeded with further screening of differentially expressed genes.

## miRNA-199b-5p has been identified as a potential target molecule in KOA

Based on the p-value and exosomal expression, combined with the results of our additional experiments (*Figure 2—figure supplement 1*), five human miRNAs (hsa-mir-3168, hsa-mir-1296–5 p, hsa-mir-15b-3p, hsa-mir-338–3 p and hsa-mir-199b-5p) were selected to further research and validated in independent human samples by RT-qPCR (hsa-mir-199b-5p, p<0.01; hsa-mir-3168, p=0.33; hsa-mir-15b-3p, p<0.01; hsa-mir-338–3 p, p=0.20 and hsa-mir-1296–5 p, p<0.01) (*Figure 2C*). To further explore the functional roles of these miRNAs, we established a mouse model of KOA to evaluate their expression in the joints. Due to species differences, the corresponding sequence of mir-1296–5 p could not be identified in the mouse model. we examined the expression of miR-199b-5p (p<0.01), miR-15b-3p (p<0.05), and miR-338–3 p (p<0.05) in mouse joint tissue samples (control *vs.* M-0.5) (*Figure 2D*). The results showed that only the expression trend of miR-199b-5p was consistent between the clinical samples and the mouse arthritis model. Therefore, we selected miR-199b-5p as the target for our subsequent research.

## MiR-199b-5p mimic inhibits the cell viability and extracellular matrix (ECM) of chondrocytes while the inhibitor restores LPS-induced chondrocytes damage

We first extracted mouse primary chondrocytes (*Figure 3A–C*). In vitro experiments showed that overexpression miR-199b-5p inhibited the viability of chondrocytes (p<0.01) (*Figure 3D*, *Figure 3—figure supplement 1*, *Figure 3—figure supplement 2*). we also find miR-199b-5p mimic increased the mRNA expressions of *MMP3* (=0.09) and *ADAMTS5* (p<0.05) and decreased the mRNA expression of *COL2A1*(p=0.05), *AGGRECAN* (p=0.20), and *SOX9* (p=0.22), which are often used as the biomarkers of chondrocytes ECM metabolic balance. In contrast, miR-199b-5p inhibitor decreased the mRNA expression of *MMP3* (p<0.01) and *ADAMTS5* (p=0.11) and increased the mRNA expression of *COL2A1* (p=0.07), *AGGRECAN* (p<0.01), and *SOX9* (p<0.01) (*Figure 3E–I*). While some gene expression changes may not be significant in statistics, but the modulation of miR-199b-5p expression has been observed to exert an influence on the metabolic alterations of chondrocytes.

To explore the effect of miR-199b-5p under pathological conditions, an LPS-stimulated inflammation chondrocyte cell model was established. We examined the effect of LPS at 5, 10, and 15 μg/ml on

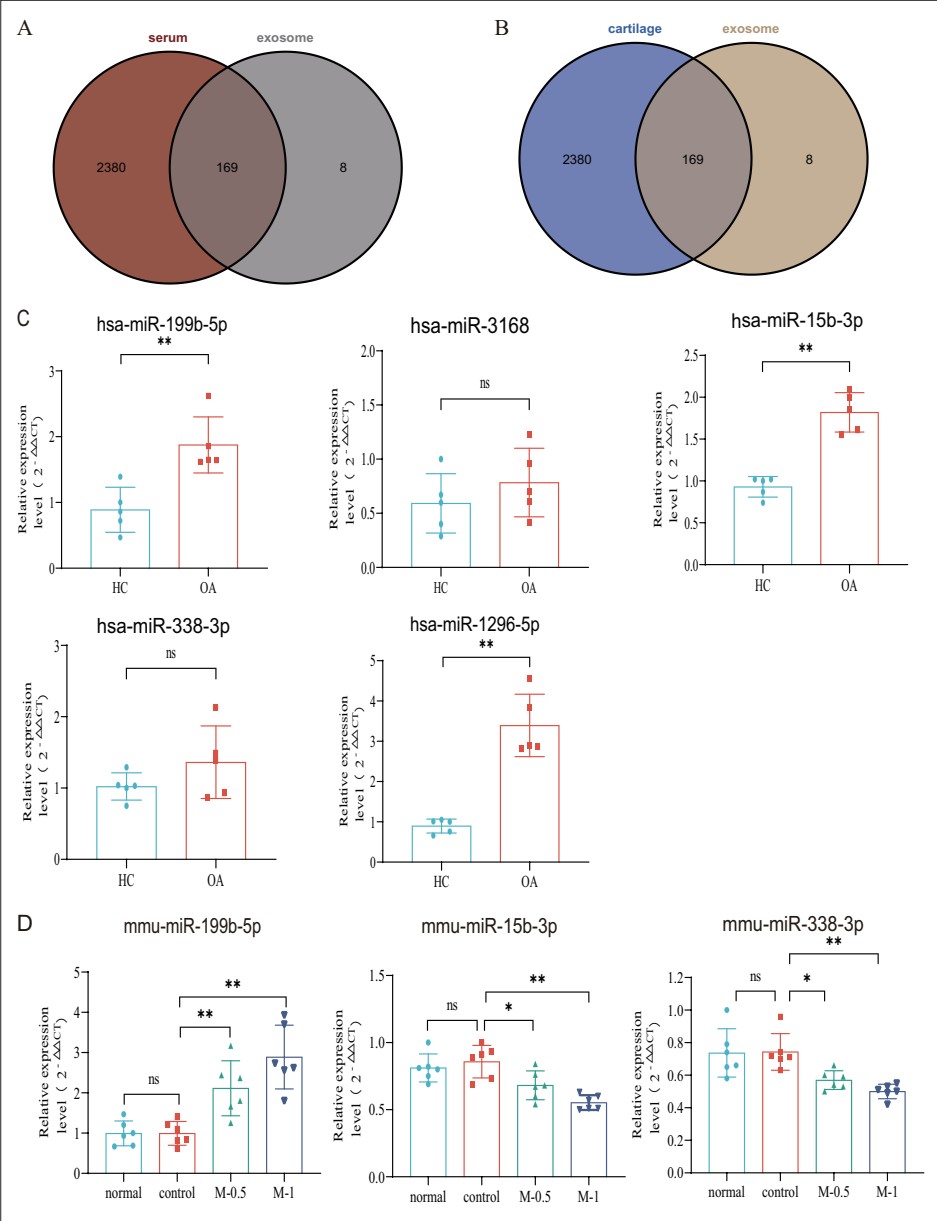

**Figure 2.** Verification with GEO data and microRNA (miRNA) screening. (**A, B**) Venn plot showing GEO dataset and our result. (**C**) RT-qPCR in clinical samples to verify the expression (HC, n=5, KOA, n=5). (**D**) RT-qPCR results of mouse joint samples (n=6). M-0.5, 10 µL of 0.5 mg/mL; M-1, 10 µL of 1 mg/mL. Data are shown as means ± SD. *p<0.05, **p<0.01.

The online version of this article includes the following source data and figure supplement(s) for figure 2:

**Source data 1.** Original data of RT-qPCR in *Figure 2C and D*.

**Figure supplement 1.** Venn diagram showing differentially expressed microRNAs (miRNAs) in the osteoarthritis (OA) group compared with healthy patients and patients who recovered after acupuncture treatment.

---

cell viability and found that cell viability was significantly decreased at 15 µg/ml(p<0.001) (*Figure 2J*). Next, we chose 15 µg/ml of LPS to establish a chondrocyte injury model. The CCK-8 assay showed that miR-199b-5p inhibitor reversed the decrease of cell viability caused by LPS (p<0.01) (*Figure 3K*). Also, we revealed that LPS elevated *MMP3* (p<0.01) and *ADAMTS5* (p<0.01) and decreased *COL2A1* (p=0.06), *AGGRECAN* (p<0.01), and *SOX9* (p=0.13) expression. In the presence of miR-199b-5p inhibitor, the changes in mRNA levels were reversed (*Figure 3L–P*). These results suggest that miR-199b-5p

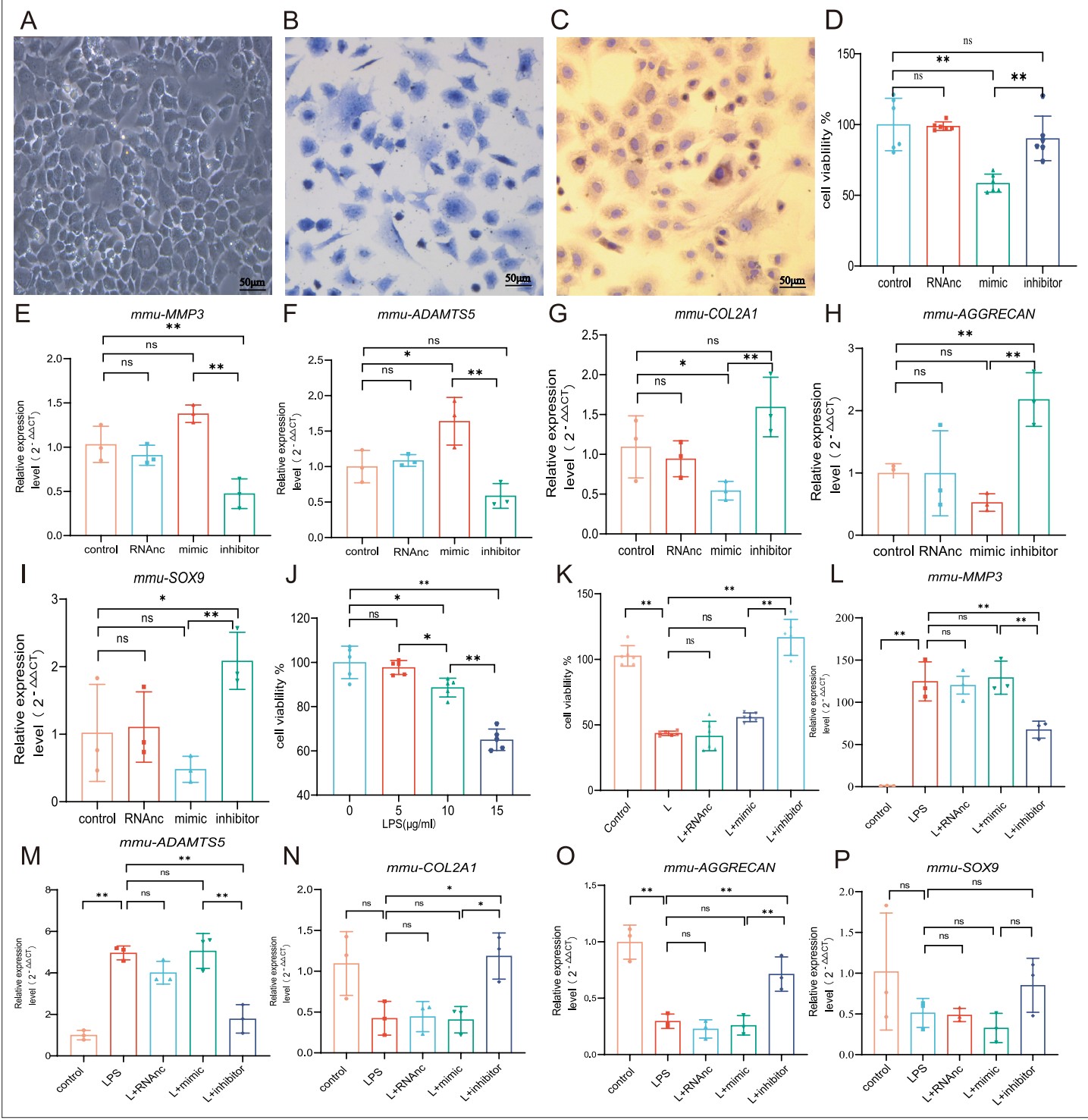

**Figure 3.** Chondrocyte proliferation and marker expression changes after miR-199b-5p mimic or inhibitor treatment. (**A**) Second-generation primary mouse chondrocytes. (**B**) Toluene blue staining. (**C**) Type II collagen immunoassay. Scale bars, 50 μm. (**D**) CCK-8 assay for cell viability (n=6). (**E, F**) RT-qPCR detection of *MMP-3* and *ADAMTS5* mRNA expression (n=3). (**G–I**) RT-qPCR detection of *COL-2A1*, *AGGRECAN*, and *SOX9* mRNA expression (n=3). (**J**) CCK-8 cell viability assay after different doses of LPS induction (n=5). (**K**) CCK-8 cell viability assay after virus infection (n=6). (**L, M**) RT-qPCR detection of *MMP-3* and *ADAMTS5* mRNA expression (n=3). (**N–P**) RT-qPCR detection of *COL2A1*, *AGGRECAN*, and *SOX9* mRNA expression (n=3). Data are shown as mean ± SD. *p<0.05, **p<0.01.

The online version of this article includes the following source data and figure supplement(s) for figure 3:

**Source data 1.** Original data of RT-qPCR in *Figure 3D–P*.

*Figure 3 continued on next page*

*Figure 3 continued*

**Figure supplement 1.** Fluorescence of adenovirus-infected chondrocytes.

**Figure supplement 2.** Primary mice chondrocytes we cultured (P1) and the secondary generation cells (P2) we used in the following experiment.

overexpression reduces cell viability and miR-199b-5p inhibition partly restores LPS-induced cell damage and ECM degradation.

## miR-199b-5p mimic induces inflammation in normal mice, and miR-199b-5p inhibitor alleviates symptoms in KOA mice

Now, we want to know the vivo role of miRNA-199b-5p. First, we screened Adneovirus (AD) and utilized High-AD as a vector to either overexpress or inhibit the expression of miR-199b-5p (***Figure 4—figure supplement 1***). In normal mice, miR-199b-5p mimic was injected into the joint of mice (***Figure 4A***) and a decrease in pain threshold was found (p<0.01) (***Figure 4B***). Four weeks later, serum *IFN-γ* (p<0.01) and *TNF-α* (p<0.01) were also significantly increased (***Figure 4C and D***). The injection of miR-199b-5p mimic in mouse joints resulted in a slight decrease in Safranine staining, although it did not reach statistical significance (***Figure 4E and F***). Additionally, the articular surface μCT image showed slight erosion (***Figure 4G***). interestingly, the level of serum inflammation in the miR-199b-5p inhibitor injection group was significantly decreased compared with the mimic group (*IFNG*, p<0.01; *TNFA*, p<0.01). These results indicated that intra-articular injection of miR-199b-5p mimic induced inflammation response in normal mice.

Followingly, we established an MIA-induced KOA model to further investigate the role of miR-199b-5p (***Figure 5—figure supplement 1***). We observed a decrease in pain threshold in the model group, and recovery was observed on the 10th day in the inhibitor group (p<0.01) (***Figure 5A and B***). Joint tissues were taken at the fourth week, and revealed that the expression of *IFNG* (p<0.01) and *TNFA* (p<0.01) decreased after inhibiting miRNA-199b-5p (***Figure 5C and D***). Safranin-fast green staining of joints showed recovery of articular cartilage degradation (p<0.05) (***Figure 5E and F***), μCT image demonstrated a partial amelioration of the cartilage erosion. (***Figure 5G***). These results proved that intra-articular injection of the miR-199b-5p inhibitor partly recovered pain, inflammation, and cartilage degeneration in KOA mice.

## *Gcnt2* and *Fzd6* are two potential target genes of miR-199b-5p

In order to investigate the underlying mechanism of miRNA-199b-5p, we utilized five widely used miRNA target gene prediction tools, namely miRWalk, miRDB, TarBase, starbase, and TargetScan, to identify potential target genes. Consequently, we identified six putative target genes of miRNA-199b-5p (***Figure 6A***). Bioinformatics analysis of the six possible target genes showed that BP is in posttranscriptional regulation of gene expression and angiogenesis; CC is in cytoplasmic vesicle and cell leading edge; and MF is in ubiquitin protein ligase binding and protein domain specific binding (***Figure 6B***).

After Mimic infected cells, we found decreased expression in *Fzd6* (p<0.01), *Gcnt2* (p<0.01), and *Caprin1* (p=0.024) in the chondrocytes (***Figure 6C***). Conversely, after inhibitor infection, *Hif1a* (p=0.048), *Fzd6* (p<0.01), and *Gcnt2* (p<0.01) were increased and *Atg14* (p=0.042) increased (***Figure 6D***). Notably, we observed corresponding changes in the expression of *Fzd6* and *Gcnt2* upon miR-199b-5p overexpression and under-expression. Moreover, we predicted potential binding sites for miRNA-199b-5p within the 3'-untranslated region (UTR) of these two target genes (***Figure 6E and F***) and luciferase reporter assays confirmed that miR-199b-5p can potentially bind to and suppress the expression of both *Gcnt2* (p<0.01) and *Fzd6* (p<0.01) *via* their complementary sequences (***Figure 6G and H***). Additionally, we carried out the comparative analysis of sequence conservatism between human and mouse, and find the binding site on 3'UTR matches to human sequence very well. Besides being positionally conserved, the sequence conservation between hsa_miR-199b-5p and mmu_miR-199b-5p was as high as 95.65%, indicating a good sequence conservation (***Figure 6—figure supplement 1***).

Furthermore, we found differential expression of *Fzd6* in both synovial tissue data (GDS5401) and chondrocyte data (GDS3758) from KOA patients in the GEO profile (***Figure 6I and J***). Similarly, the differential expression tendency of *Gcnt2* was observed in both the synovial tissue data (GDS5403)

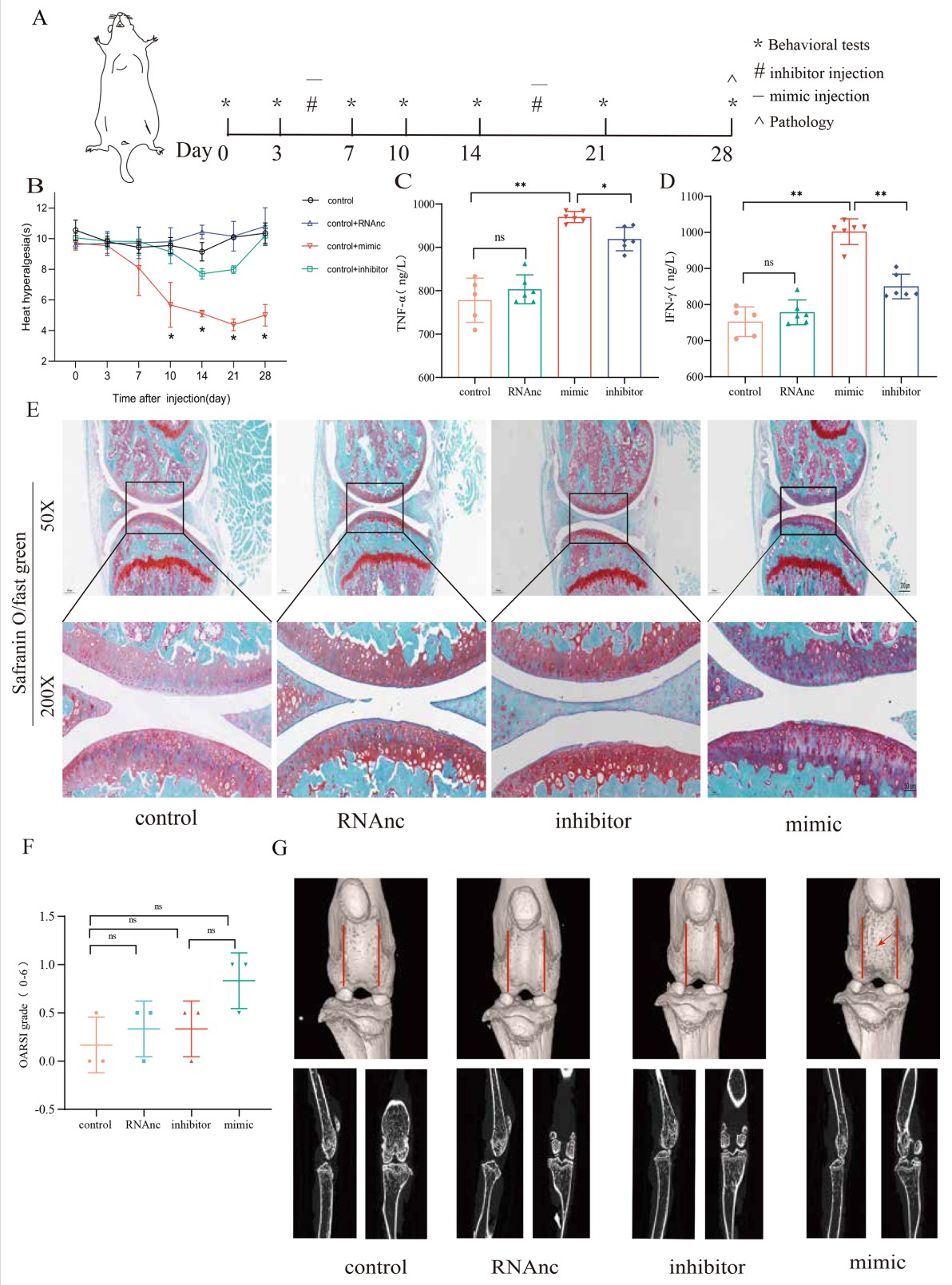

**Figure 4.** Injection of adenovirus expressing miR-199b-5p mimic results in inflammation and pain threshold sensitivity in mice. (**A**) Animal experiment schematic. (**B**) Behavioral detection of animal thermal pain threshold. (**C, D**) Detection of serum levels of *IFN-γ* and *TNF-α* in controls by ELISA. (**E, F**) Safranin-fast green staining and semiquantitative scoring of articular cartilage. Scale bar = 200 μm (top), 50 μm (bottom). (**G**) 3D reconstruction and 2D images of joints from μCT scans. Data are shown as mean ± SD. *p<0.05, **p<0.01, n=6.

*Figure 4 continued on next page*

*Figure 4 continued*

The online version of this article includes the following source data and figure supplement(s) for figure 4:

**Source data 1.** Thermal pain of the original data in *Figure 4B*; ELISA of the original data in *Figure 4C and D*; OARSI scores of the original data in *Figure 4F*.

**Figure supplement 1.** Expression of adenovirus in mouse knee joint.

and GDS3758 from the same KOA patients (*Figure 6K and L*). These findings provide further validation to our results. Therefore, *Fzd6* and *Gcnt2* was considered by us as an important downstream target for further research.

## Discussion

In this study, we initially performed sequencing of serum exosomal miRNAs from clinical patients and identified 177 dysregulated miRNAs. Subsequently, through comparison with GEO data, we found that 169 miRNAs were expressed in both KOA serum and cartilage. Following a screening process, miR-199b-5p was selected for further experiments. In cell-based assays, we discovered that miR-199b-5p can influence the viability of chondrocytes and cytokine-mediated extracellular matrix metabolism. Moreover, in vitro experiments demonstrated that it can induce inflammation and abnormal pain threshold in normal mice. Importantly, inhibition of miR-199b-5p alleviated the pathological symptoms of KOA. Finally, these effects were achieved by potentially targeting *Gcnt2* and *Fzd6* (*Figure 7*). Thus, our findings demonstrated that miR-199b-5p might be a novel potential therapeutic target for OA prevention and treatment.

miRNAs function as regulators of gene expression in biological processes by regulating mRNA translation by specifically binding to the 3'UTRs of target mRNAs (*Bartel, 2004*). Exosomes are rich in miRNAs, and various cells can secrete exosomes to target cells under physiological and pathological conditions, which function as delivery vehicles for miRNAs (*Kalluri and LeBleu, 2020*; *Sun et al., 2019*). There is evidence suggesting that miRNAs can participate in various cellular processes (inflammation, cell viability, ECM dysregulation) and signaling pathways (Hedgehog signaling, PI3K-AKT signaling) relevant to OA (15, *Ali et al., 2021*; *Jiang et al., 2023*). Our differential miRNA enrichment analysis also strongly supports the correlation with these findings.

In cancer research, overexpression of miR-199b-5p can inhibit the proliferation, migration, and invasion of prostate cancer cells in vitro and tumor growth and metastasis in vivo by targeting DDR1 (*Zhao et al., 2021*). In addition, exogenous miR-199b-5p inhibited the growth of bone marrow mesenchymal stem cells (MSCs) and promoted the differentiation of bone MSCs into chondrocytes by targeting the *JAG1* pathway (*Zhang et al., 2020*; *Qu et al., 2017*). Our functional experiments showed that overexpression of miR-199b-5p reduced chondrocyte viability and the expression of anabolic factors such as COL2A1, AGGREGN, and SOX9. It also increased inflammation levels and decreased pain thresholds in control mice. Although no pathological changes were observed in the articular cartilage of control mice, a similar study demonstrated that pathological cartilage changes only occurred after six months in mice with miR-211 and miR-204 knockout (*Huang et al., 2019*). Therefore, we speculate that the time of overexpression of miR-199b-5p in our experiment was too short, so it only caused inflammation and pain threshold response. To the best of our knowledge, this is the initial investigation documenting the involvement of miR-199b-5p in KOA.

*Fzd6* is known to be up-regulated during the osteogenic differentiation of MSCs and can be regulated by miR-194–5 p to activate the WNT signaling pathway, thereby promoting osteogenic differentiation of MSCs (*Lu et al., 2022*). *Gcnt2* has been found to induce epithelial-mesenchymal transition and enhance migration and invasion of esophageal squamous cell carcinoma cells (*Peng et al., 2019*). Our findings indicate that miR-199b-5p plays a crucial role in KOA by potentially targeting *Fzd6* and *Gcnt2*. However, whether miR-199b-5p truly functions through *Fzd6* and/or *Gcnt2* requires genetic knockdown of *Fzd6* and *Gcnt2* in the presence of miR-199b-5p. Future research should further explore the roles and mechanisms of *Fzd6* and *Gcnt2* in the context of KOA.

This study also has some limitations. We initially detected serum exosomes miRNA and later examined the miRNA through in vitro experiment and animal study. However, whether the miRNAs targeting the joints have the same role as the serum exosome miRNAs or blood miRNA has not been clear. We also performed direct intra-articular injection of adenoviral vectors. The intra-articular

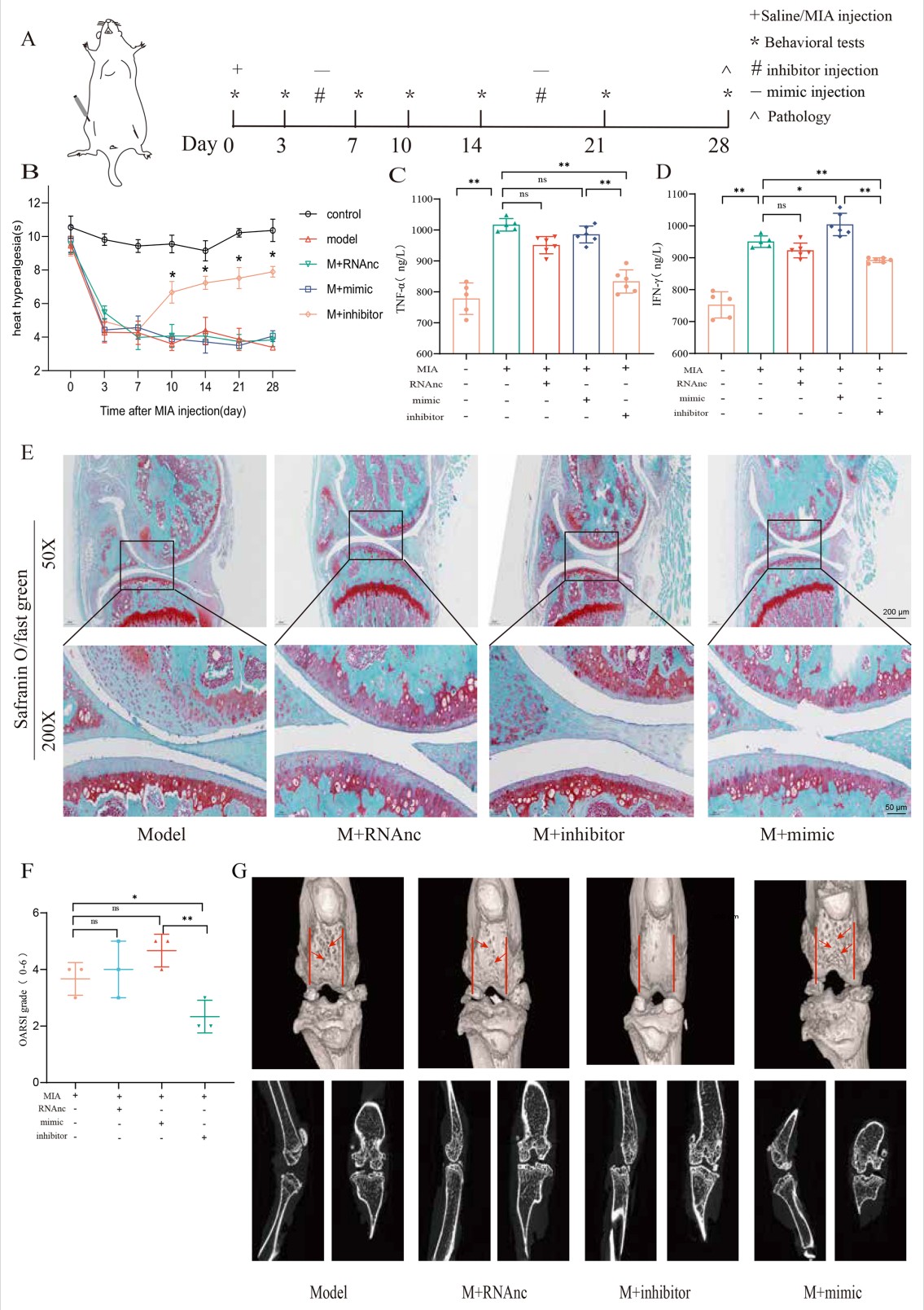

**Figure 5.** Injection of adenovirus expressing miR-199b-5p inhibitor partly recovered pathological changes in knee osteoarthritis (KOA) mice. (**A**) Animal experiment schematic. (**B**) Behavioral detection of animal thermal pain threshold. (**C, D**) Detection of serum levels of *IFN-γ* and *TNF-α* in controls by ELISA. (**E, F**) Safranin-fast green staining and semiquantitative scoring of articular cartilage. Scale bar = 200 μm (top), 50 μm (bottom) (**G**) 3D reconstruction and 2D images of joints from μCT scans. Data are shown as mean ± SD. *p<0.05, **p<0.01, n=6.

*Figure 5 continued on next page*

*Figure 5 continued*

The online version of this article includes the following source data and figure supplement(s) for figure 5:

**Source data 1.** Thermal pain of the original data in *Figure 5B*; ELISA of the original data in *Figure 5C and D*; OARSI scores of the original data in *Figure 5F*.

**Figure supplement 1.** Establishment of knee osteoarthritis (KOA) model in mice by injection of MIA.

**Figure supplement 1—source data 1.** Thermal pain of the original data in *Figure 5—figure supplement 1A*; OARSI scores of the original data in *Figure 5—figure supplement 1D*.

injection can reduce the exposure of extra-articular tissue, thereby minimizing the associated side effects and improving targeting. In this study, the chondrocyte experiments were conducted in a 2D manner, which may lead to chondrocyte de-differentiation and thus may weaken the conclusion of the chondrocyte response to the treatments. Therefore, in the future study, we will adopt 3D culture system for experiments. Besides, more in vivo study such as gene knockout mice study can be used in the future study.

In conclusion, we found that miR-199b-5p is elevated in osteoarthritis and may affect cell viability and related cytokines by potentially targeting *Gcnt2* and *Fzd6*. Overexpression of miR-199b-5p induced OA-like pathological changes in normal mice and inhibiting miR-199b-5p alleviated symptoms in KOA mice, suggesting it may be a target for treatment of OA. Through human clinical trials, cell experiments, and animal models, we not only identified a new OA-related miR-199b-5p but also examined the biological function of miR-199b-5p in OA.

## Materials and methods

### Human samples

Study participants were recruited from the Hospital of Chengdu University of Traditional Chinese Medicine and the surrounding communities. The study was approved by the Ethics Review Committee of the Hospital at Chengdu University of Traditional Chinese Medicine (2016KL-017) and conformed to the ethical guidelines of the 1975 Declaration of Helsinki. Serum was collected from all participants and serum samples were stored at –80 °C.

### Human subjects

Patients were enrolled if they fulfilled the following criteria: (1) diagnosed with KOA according to the ACR; (2) aged between 40 and 65 years; (3) agreed to cooperate with researchers in all research procedures after enrollment; (4) provided with written informed consent. Patients with any of the following conditions were excluded: (1) accompanied with other diseases such as rheumatoid arthritis, bone tumors, and bone tuberculosis; (2) treated with intra-articular glucocorticoid or viscoelastic supplementation in the last six months; (3) knee replacement history; (4) had complicated cardiovascular disease, diabetes, skin disease, and liver or kidney impairment; (5) pregnant or breastfeeding women; (6) accompanied with mental and intellectual disabilities; or (7) undergoing other clinical trials.

### Extraction and sequencing of serum exosome miRNAs

Serum samples were filtered using 0.22 µM filters. Exosomes were isolated from the serum sample using ExoQuick Exosome Precipitation Solution (System Biosciences) following the manufacturer's instructions. Briefly, serum was thawed on ice and centrifuged at 4000 rpm for 15 min to remove any cells or cellular debris. Next, 50 µL ExoQuick Solution was added into the 200 µL serum sample and mixed thoroughly. The exosomes were suspended in PBS.

Exosomes were characterized by electron microscopy (Tecnai G2 Spirit 120KV, FEI), nanoparticle tracking analysis (NTA) (NTA 3.2 Dev Build 3.2.16), and western blot analysis (for CD9, CD63, and CD81). Sequencing was performed by single-end sequencing (1×150 bp) on Illumina NextSeq 500. The libraries were sequenced on an Agilent 2100 Bioanalyzer platform. The mirdeep2 software (https://www.mdc-berlin.de/content/mirdeep2-documentation) was used to analyze the miRNA sequences and quantification. The heatmap was plotted based on the log2 (fold change), using Heatmap Illustrator software (Heml 1.0).

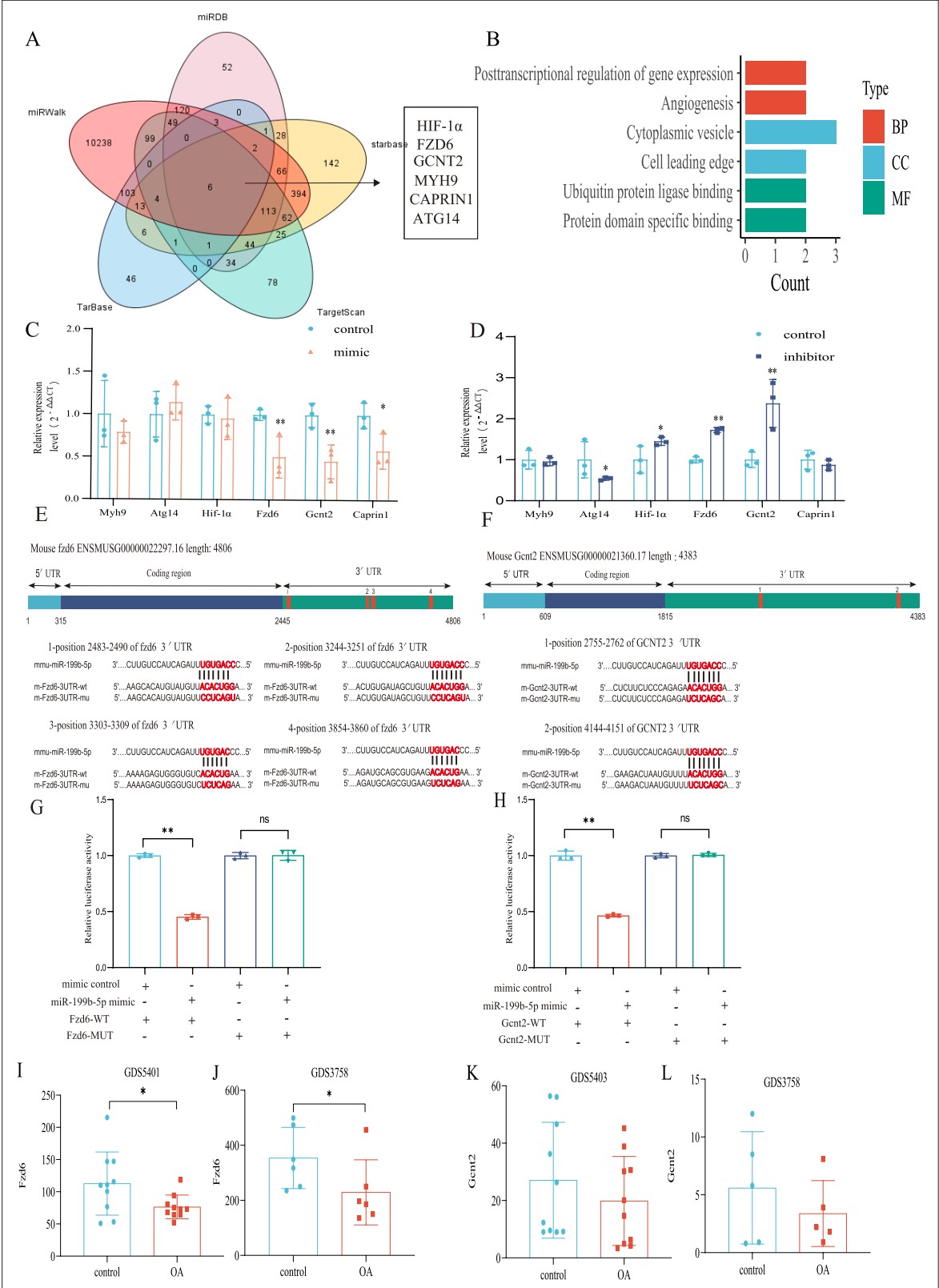

**Figure 6.** Validation of the miR-199b-5p target gene. (**A**) Prediction of target genes of miR-199b-5p using search sites. (**B**) Target gene gene ontology (GO) analysis. (**C, D**) Detection of the expression of target genes under different conditions (n=3). (**E, F**) We use targetscan to predict the binding site of miR-199b-5p and target genes. (**G, H**) Validation by luciferase reporter gene assay (n=3). Data are show as mean ± SD.*p<0.05, **p<0.01, n=6. (**I–L**) The expression of *Fzd6* and *Gcnt2* in the synovial membrane and chondrocytes of GEO Profiles Knee OA (KOA).

*Figure 6 continued on next page*

*Figure 6 continued*

The online version of this article includes the following source data and figure supplement(s) for figure 6:

**Source data 1.** Original data of RT-qPCR in *Figure 6C and D*; Original data of luciferase assay in *Figure 6G and H*; Original data of GDS in *Figure 6I–L*.

**Source data 2.** We predicted the potential binding sites of miRNA-199b-5p in the 3'-untranslated regions (UTRs) of two target genes, *Fzd6* and *Gcnt2*, in both human and mouse.

**Figure supplement 1.** Comparative analysis of sequence conservatism between human and mouse.

## Total RNA was isolated from human serum, cell, and mice

Chondrocytes using a kit (Yeasen, Shanghai, China) according to the manufacturer's instructions. Reverse transcription was performed using 1000 ng total RNA and a Prime Script RT Reagent Kit (Yeasen, Shanghai, China) or Prime Script RT Master Mix (Yeasen, Shanghai, China), which were used for miRNA and mRNA, respectively. For miRNA, the reactions were incubated at 42 °C for 15 min followed by inactivation at 85 °C for 5 s. qRT-PCR amplification was assessed in a CFX Connection Real-Time System (Bio-Rad) using the SYBR Premix Ex Taq II kit (Yeasen, Shanghai, China). The following cycling conditions were used: 95 °C for 30 s, followed by 40 cycles of 95 °C for 5 s and 60 °C for 30 s. All reactions were performed in duplicate and normalized to the internal reference U6 for miRNA and GAPDH mRNA for mRNAs. The $2^{-\Delta\Delta CT}$ CT method was used to evaluate the relative mRNA/miRNA expression levels.

## RNA primer

Primers are listed in *Supplementary file 1*, Supplement table 1.

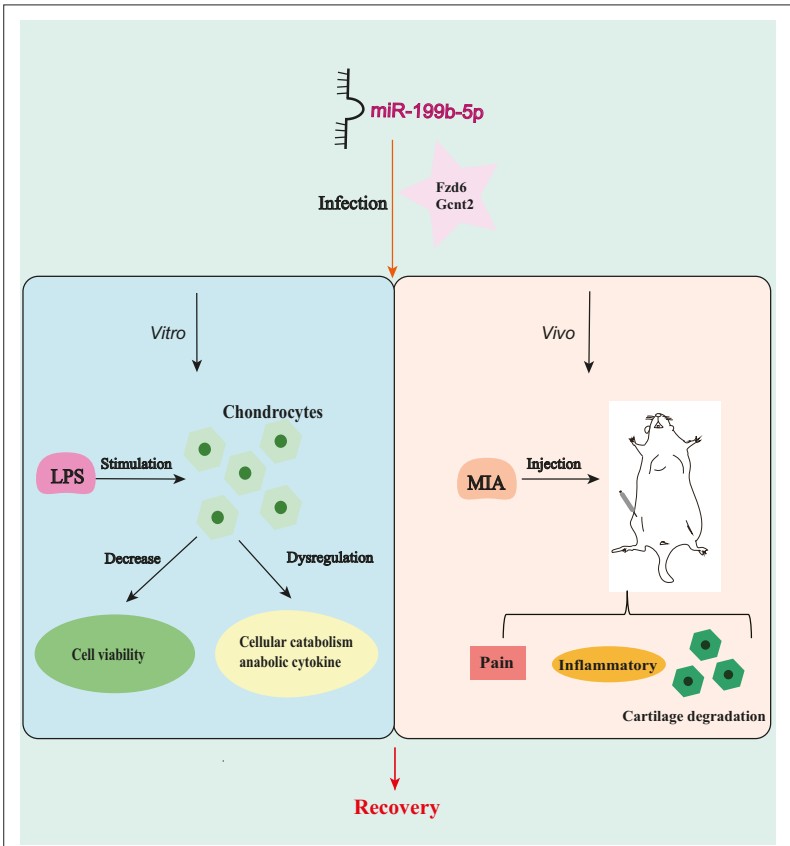

**Figure 7.** miR-199b-5p exerts its effects on in vitro cells and in vivo mice by potentially targeting *Fzd6* and *Gcnt2*.

## Bioinformatics analysis

GO analysis was performed, involving the biological process (BP), cellular component (CC), and molecular function (MF), using DAVID (*Sherman et al., 2022*) (https://david.ncifcrf.gov/home.jsp). We used the miRNA target gene prediction websites miRDB (*Chen and Wang, 2020*) (http://mirdb.org/), miRWalk (*Dweep et al., 2011*) (http://mirwalk.umm.uni-heidelberg.de/), Starbase (*Li et al., 2014*) (https://starbase.sysu.edu.cn/), DIANA-TarBase (*Karagkouni et al., 2018*) (http://diana.imis.athena-innovation.gr/DianaTools/ index.php?*r*=tarbase/index), Targetscan (*Agarwal et al., 2015*) (https://www.targetscan.org/vert_72/). The target genes of mmu-miR-199b-5p were predicted, and the intersection of the target gene prediction list results of the five sites was taken.

The dysregulated miRNAs were compared to relevant published miRNA data from human KOA patients. The dataset GSE105027 was obtained from serum samples of KOA patients, while GSE175961 comprised sequencing data from KOA patient cartilage. Finally, we validated the expression of the target genes Gcnt2 and Fzd6 in KOA patients using synovial data (GDS5401, GDS5403) and cartilage data (GDS3758). All operations will be performed using the R software.

## Primary mouse chondrocyte culture and other cell experiments

According to the protocol by Gosset et al (*Pitcher et al., 2016*). young mice 3–5 days after birth were taken, cartilage tissues were extracted, and the cells were cultured in the complete medium after digestion with collagenase. The cells were cultured in F12 medium containing 10% fetal bovine serum and 1% penicillin-streptomycin at 37 °C and 5% $CO_2$. The medium was changed every three days, and cells were cultured to the second passage for experiments. Throughout the experiment, we used second-generation chondrocytes and measured cell viability using the CCK-8 assay. The stimulation time for lipopolysaccharide-induced inflammatory damage model was 6 hr. Regarding virus infection, the cells were infected after 36 hr of passage when their density reached approximately 30%. According to the virus operation manual, the total infection time was approximately 40 hr.

## Animal model of KOA

Animal experiments were performed using a total of 108 8-week-old male C57BL/6 mice. The protocol was approved by the Committee on the Ethics of Animal Experiments of Chengdu University of Traditional Chinese Medicine (2019–04). After randomization, the KOA mouse model was established by orthotopic injection of MIA (sigma) into the knee joint of mice. After the animals were anesthetized with isoflurane in a small animal anesthesia machine, the right knee joint of the mouse was shaved, the knee joint was flexed 90°, and 10 µL of MIA solution was injected into the knee joint cavity using a 28 G microsyringe. The syringe needle was slowly withdrawn and the knee joint was gently moved. Baseline testing of behavioral pain thresholds was conducted prior to model establishment. After the model was established, the behavioral pain threshold testing was on days 3, 7, 10, 14, 21, and 28 after model establishment. After the 3rd day and 14th day after the start of the experiment, the experimental mice were injected with adenovirus (total volume 10 µL) in the knee joint, and the control mice were injected with empty adenovirus as a control (Hanbio tech, Shanghai, China) (*Choi et al., 2019*).

## Thermal pain threshold detection

A pain threshold detection instrument was used to measure the latency of the right plantar leg raising reflex of mice under heat radiation. Three days before the experiment, mice were cut off from water for half a day and adapted to the environment for 1 hr. A certain intensity of pyrogen was used to irradiate the plantar position of the mouse's right limb, and a machine was used to automatically record the time when the mouse moved the limb from thermal pain. The resting intensity of the thermal pain stimulator was set to 10%, and the maximum duration was set to 20 s to avoid prolonged heating and burning of skin. After the light source is aimed at the sole of the mouse, we observed the mouse in real-time. If paw raising, paw licking, or paw retraction was observed, the irradiation was stopped and the irradiation time of the instrument was recorded. The right limb of each mouse was tested five times, with an interval of 10 min each time; outliers were eliminated, and the average value was calculated and included in the statistics (*Cheah et al., 2017*).

## ELISA and μCT scanning

Mice were sacrificed after four weeks, and serum was collected for ELISA assay (Jiangsu Jingmei Biotechnology Co., Ltd., Yancheng, China). The knee joint specimens of the right hindlimb of mice were scanned using a high-resolution micro-CT skysan1267 instrument (Bruker, Germany). The sample was removed from fixative and dried. The scanning parameters were as follows: voltage 55 KV, current 200 μA, and filter 0.25AL. After scanning, three-dimensional reconstruction was performed using NRecon1.7.4.2 software.

## Safranin fast green staining

Mouse knee joint staining was performed using the Safranin Fast Green Staining Kit (Servicebio Biological Technology, Wuhan, China). The sample was examined under a microscope, and the cartilage integrity was scored according to the OARSI grading system, in which the score ranged from 0 to 6 points (*Glasson et al., 2010*).

## Luciferase assay

We utilized Targetscan to predict potential binding sites and designed sequences accordingly. The wild-type (WT) and mutant-type (MUT) sequences (according to the predicted binding site) were inserted into the pmiRGLO plasmid. HEK-293T cells were seeded in six-well plates at 24 hr before transfection. GCNT2 and FZD6 3′UTR-wt and gcnt2 and fzd6 3′UTR-mut plasmids (500 ng) and 20 nmol miR-199b-3p and NC were co-transfected with Lipofectamine 3000 (Hanbio Biotechnology, China) following the manufacturer's instructions. After 48 hr, firefly and Renilla luciferase activities were calculated using the Promega Dual-Luciferase system following the manufacturer's instructions (Hanbio tech, Shanghai, China). Firefly/Renilla luciferase was measured to evaluate relative luciferase activity.

## Comparative analysis of sequence conservatism between human and mouse

First, the sequence information of mmu_miRNA-199b-5p was used to locate the human homologous sequence in the UCSC (*Raney et al., 2024*) database. Based on this positional information and the source gene, a further comparison was conducted in miRbase (*Kozomara et al., 2019*) to identify the nearest miRNA at the position of the human genome.

## Statistical analysis

Data are reported as the mean ± SD. Normal distribution and homogeneity of variance of data were first tested. Shapiro-Wilk test was used to verify data normality, while Levene's test or Browne-Forsythe test was adopted for the assessment of variance equality. Statistical analysis was performed by unpaired two *t*-tests and Tukey-corrected one-way analysis of variance (ANOVA) for comparisons between groups. Tukey-corrected two-way ANOVA for the comparison of mice thermal pain threshold data. $p < 0.05$ was considered statistically significant for all statistical calculations. PRISM 8.0 (GraphPad Software, San Diego, CA, USA) was used for data analysis.

## Acknowledgements

This work was supported by the National Key R&D Program of China (No. 2019YFC1709000, 2022YFC3500703); Fund of Science and Technology Department of Sichuan Province, China (No. 2022ZDZX0033); and Xinglin Scholar Foundation of Chengdu University of Traditional Chinese Medicine (No. QNTD2022003). We also grateful to Zheng-jie Li, Xiao-hua Peng, and Ke-li Zhu for good advices.

## Additional information

### Funding

| Funder | Grant reference number | Author |
|---|---|---|
| National Key Research and Development Program of China | 2019YFC1709000 | Shu-Guang Yu |
| National Key Research and Development Program of China | 2022YFC3500703 | Qiao-Feng Wu |
| Fund of Science and Technology Department of Sichuan Province | 2022ZDZX0033 | Qiao-Feng Wu |
| Xinglin Scholar Foundation of Chengdu University of Traditional Chinese Medicine | QNTD2022003 | Qiao-Feng Wu |

The funders had no role in study design, data collection and interpretation, or the decision to submit the work for publication.

### Author contributions

Tong Feng, Conceptualization, Visualization, Writing – original draft, Writing – review and editing; Qi Zhang, Si-Hui Li, Yan-ling Ping, Mu-qiu Tian, Jun-Meng Wang, Writing - review and editing; Shuan-hu Zhou, Xin Wang, Fan-Rang Liang, Shu-Guang Yu, Data curation; Qiao-Feng Wu, Conceptualization, Writing – review and editing

### Author ORCIDs

Tong Feng (ID) http://orcid.org/0009-0004-5022-8422
Mu-qiu Tian (ID) http://orcid.org/0009-0009-3142-7270
Qiao-Feng Wu (ID) https://orcid.org/0000-0002-8870-6707

### Ethics

Clinical trial registration 2016KL-017.

Study participants were recruited from the Hospital of Chengdu University of Traditional Chinese Medicine and the surrounding communities with written informed consent. The study was approved by the Ethics Review Committee of the Hospital at Chengdu University of Traditional Chinese Medicine (2016KL-017) and conformed to the ethical guidelines of the 1975 Declaration of Helsinki. Animal experiments were performed using total of 108 8-week-old male C57BL/6 mice. The protocol was approved by the Committee on the Ethics of Animal Experiments of Chengdu University of Traditional Chinese Medicine (2019-04).

Reviewer #1 (Public Review): https://doi.org/10.7554/eLife.92645.3.sa1
Reviewer #2 (Public Review): https://doi.org/10.7554/eLife.92645.3.sa2
Author response https://doi.org/10.7554/eLife.92645.3.sa3

## Additional files

### Supplementary files

• Supplementary file 1. Supplement table 1: Primers list; Supplement table 2: Basic information of participants.

• Source data 1. Original data of Basic information for recruiting patients is in *Supplementary file 1*, Supplement table 2.

• MDAR checklist

## Data availability

Sequencing data have been deposited in GEO under accession code GSE263996. All data are available in the main text or the supplementary materials.

The following dataset was generated:

| Author(s) | Year | Dataset title | Dataset URL | Database and Identifier |
|---|---|---|---|---|
| Feng T, Zhang Q | 2024 | Exosomal miRNAs from serum sequencing | https://www.ncbi.nlm.nih.gov/geo/query/acc.cgi?acc=GSE263996 | NCBI Gene Expression Omnibus, GSE263996 |

The following previously published datasets were used:

| Author(s) | Year | Dataset title | Dataset URL | Database and Identifier |
|---|---|---|---|---|
| Ntoumou E, Braoudaki M, Lambrou GI, Tzetis M, Tsezou A | 2017 | Circulating microRNA signature as novel biomarkers for osteoarthritis development | https://www.ncbi.nlm.nih.gov/geo/query/acc.cgi?acc=GSE105027 | NCBI Gene Expression Omnibus, GSE105027 |
| Jing J, Zeng G | 2021 | Microarray and differential expression analysis were used to identify the noncoding RNAs (ncRNAs) and mRNAs that were expressed abnormally between the cartilage from KOA patients and healthy controls (miRNA) | https://www.ncbi.nlm.nih.gov/geo/query/acc.cgi?acc=GSE175961 | NCBI Gene Expression Omnibus, GSE175961 |
| Woetzel D, Huber R, Kupfer P, Pohlers D, Pfaff M, Driesch D, Häupl T, Koczan D, Stiehl P, Guthke R, Kinne RW | 2014 | Identification of rheumatoid arthritis and osteoarthritis patients by transcriptome-based rule set generation | https://www.ncbi.nlm.nih.gov/geo/query/acc.cgi?acc=GSE55235 | NCBI Gene Expression Omnibus, GSE55235 |
| Woetzel D, Huber R, Kupfer P, Pohlers D, Pfaff M, Driesch D, Haeupl T, Koczan D, Stiehl P, Guthke R, Kinne RW | 2014 | Identification of rheumatoid arthritis and osteoarthritis patients by transcriptome-based rule set generation [Jena] | https://www.ncbi.nlm.nih.gov/geo/query/acc.cgi?acc=GSE55457 | NCBI Gene Expression Omnibus, GSE55457 |
| Dehne T, Karlsson C, Ringe J, Sittinger M, Lindahl A | 2009 | Chondrogenic differentiation potential of OA chondrocytes and their use in autologous chondrocyte transplantation | https://www.ncbi.nlm.nih.gov/geo/query/acc.cgi?acc=GSE16464 | NCBI Gene Expression Omnibus, GSE16464 |

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
