## [Editor Report · eLife assessment]

This **valuable** study reports that miR-199b-5p is elevated in human osteoarthritis patients. There is **solid** evidence for the finding that inhibiting miR-199b-5p alleviates symptoms in mice with knee osteoarthritis. Additionally, potential targets of miR-199b-5p are identified but whether miR-199b-5p truly functions through Fzd6 and/or Gcnt2 requires further investigation.

---

## [Referee Report · Reviewer #1 (Public Review)]

Summary:

In this manuscript, the authors reported that miR-199b-5p is elevated in osteoarthritis (OA) patients. They also found that overexpression of miR-199b-5p induced OA-like pathological changes in normal mice and inhibiting miR-199b-5p alleviated symptoms in knee OA mice. They concluded that miR-199b-5p is not only a potential micro target for knee OA, but also provides a potential strategy for future identification of new molecular drugs.

Strengths:

The data are generated from both human patients and animal models. The data presented in this revised manuscript is solid and support their conclusions. The questions from reviewers are also properly addressed and the quality of this manuscript has been significantly improved.

There are no significant weaknesses identified in this revised manuscript.

---

## [Referee Report · Reviewer #2 (Public Review)]

Summary:

The Authors identified miR-199b-5p is a potential OA target gene using serum exosomal small RNA-seq from human healthy and OA patients. Their RNA-seq results were further compared with publicly available datasets to validate their finding of miR-199b-5p. In vitro chondrocyte culture with miR-199b-5p mimic/inhibitor and in vivo animal models were used to evaluate the function of miR-199b-5p in OA. The possible genes that were potentially regulated by miR-199b-5p were also predicted (i.e., Fzd6 and Gcnt2) and then validated by using Luciferase assays.

Strengths:

(1) Strong in vivo animal models including pain tests.

(2) Validate the binding of miR-199b-5p with Fzd6 and binding of miR-199b-5p with Gcnt2

The authors have addressed my concerns.

---

## [Author Response]

The following is the authors’ response to the original reviews.

We thank the constructive criticism provided by the reviewers and editor. Based on these suggestions, we have thoroughly reworked the manuscript. More specifically but not limit:

(1) We have corrected the mistakes mentioned by the reviewers on a point-by-point basis.

(2) We have provided additional experimental evidences to explain the rationale behind selecting five miRNAs for q-PCR validation. Furthermore, we have elaborated on the reasons for focusing primarily on research related to cartilage.

(3) In response to concerns regarding overinterpretation in the manuscript, we have made more precise descriptions and revisions. Furthermore, we have added some details in our methods, including the addition of results showing the conservation of miR-199b-5p sequences between human and mouse species.

(4) We have provided additional details on the experiments, including the process for predicting target genes, timing of chondrocyte culture and other experimental operations.

(5) Finally, we have made additional revisions to the details of the figures to avoid any distortions and enhance the precision of the language.

Below please find our responses to the reviewers’ comments on a point-by-point basis. You also can track the changes in the modified manuscript. We believe that this revision has been substantially improved.

**eLife assessment**
The manuscript provides interesting evidence that miR-199b-5p regulates osteoarthritis and as such it may be considered as a potential therapeutic target. This finding may be useful to further advance the field.

Thank you for your positive comments.

Although the study is considered potentially clinically relevant, the evidence provided was deemed insufficient and incomplete to support the conclusions drawn by the authors.

Thank you for your critical comments and constructive advices. We have response point to point according to the reviewers’ questions and thoroughly re-working our manuscript. We hope the revised manuscript can be qualified to the criteria and be published on the journal of eLife.

**Reviewer #1 (Public Review):**
Summary:In this manuscript, the authors observed that miR-199b-5p is elevated in osteoarthritis (OA) patients. They also found that overexpression of miR-199b-5p induced OA-like pathological changes in normal mice and inhibiting miR-199b-5p alleviated symptoms in knee OA mice. They concluded that miR-199b-5p is not only a potential micro-target for knee OA but also provides a potential strategy for the future identification of new molecular drugs.

Thanks for your comment.

Strengths:The data are generated from both human patients and animal models.

Thanks for the positive comment.

Weaknesses:The data presented in this manuscript is not solid enough to support their conclusions. There are several questions that need to be addressed to improve the quality of this study.The following questions that need to be addressed to improve the quality of the study.(1) Exosomes were characterized by electron microscopy and western blot analysis (for CD9, 264 CD63, and CD81). However, figure S1 only showed two sample WB results and there is no positive and negative control as well as the confused not clear WB figure.

Thank you for your suggestion. We acknowledge that a comprehensive identification of extracellular vesicles should include both positive and negative samples. However, in some of the initial studies we referenced, the positive and negative control were not mentioned1;2. In our study, we identified extracellular vesicles using a combination of electron microscopy, nanoparticle tracking analysis, and marker detection of exosomes. We agree that having negative samples would make our results more convincing, and we will include a negative control group in our future experiments. Additionally, we have provided clearer images in the revised version. (supplemental fig1 A)

Reference

(1) Ying W, Riopel M, Bandyopadhyay G, et al. Adipose Tissue Macrophage-Derived Exosomal miRNAs Can Modulate In Vivo and In Vitro Insulin Sensitivity. Cell. 2017;171(2).

(2) Fang T, Lv H, Lv G, et al. Tumor-derived exosomal miR-1247-3p induces cancer-associated fibroblast activation to foster lung metastasis of liver cancer. Nature Communications. 2018;9(1):191.

(2) The sequencing of miRNAs in serum exosomes showed that 88 miRNAs were upregulated and 89 miRNAs were downregulated in KOA patients compared with the control group based on fold change > 1.5 and p < 0.05. Figure 2 legend did not clearly elucidate what those represent and why the authors chose those five miRNAs to further validate although they did mention it with several words in line 108 'based on the p-value and exosomal'.

In fact, our study included two additional groups: the acupuncture treatment group (4 weeks of continuous acupuncture treatment) and the waiting treatment group (no intervention, followed by acupuncture treatment after 4 weeks), in addition to the healthy control and knee osteoarthritis (OA) patient groups. After comparing these four groups, we found that 11 genes (hsa-miR-504-3p, hsa-miR-1915-3p, hsa-miR-103a-2-5p, hsa-miR-887-3p, hsa-miR-1228-5p, hsa-miR-34c-3p, hsa-miR-3168, hsa-miR-518e-3p, hsa-miR-1296-5p, hsa-miR-338-3p, and hsa-miR-199b-5p) were upregulated in KOA patients but downregulated after acupuncture treatment, with no change in the waiting treatment group. Additionally, 7 genes (hsa-miR-448, hsa-miR-514a-3p, hsa-miR-4440, hsa-let-7f-5p, hsa-let-7a-5p, hsa-let-7d-5p, and hsa-miR-15b-3p) were downregulated in KOA patients but upregulated after acupuncture treatment, with no change in the waiting treatment group. Considering the improvement in clinical symptoms of KOA patients after acupuncture treatment, we believe that these 18 genes are of significant value. Based on overall expression abundance and species specificity, we finally selected 5 genes, namely the 5 genes mentioned in this article. Regarding this result, we have already included it in the supplementary fig5(fig. S5).

**Author response image 1. sa3fig1:** Venn diagram showing differentially expressed miRNAs in the OA group compared with healthy patients and patients who recovered after acupuncture treatment.

(3) In Figure 3 legend and methods, the authors did not mention how they performed the cell viability assay. What cell had been used? How long were they treated and all the details? Other figure legends have the same problem without detailed information.

Thank you for your suggestions. In Figure 3, cell viability was determined using the CCK-8 assay. We used second-generation chondrocytes for this analysis. The chondrocytes were obtained from young mice aged 3-5 days after birth. The cartilage tissues were extracted, and the cells were cultured in complete medium after digestion with collagenase. The detailed description of the cell viability assay, cell culture procedures, specific timing, and treatment methods of the cells used can be found in our revised manuscript. (page14-15，line304-313)

Besides, we have made thorough revisions to all figure legends to provide a clearer explanation of the relevant content.

(4) The authors claimed that Gcnt2 and Fzd6 are two target genes of miR-199b-5p. However, there is no convincing evidence such as western blot to support their bioinformatics prediction.

In the current study, we first identified six potential target genes by intersecting the predicted targets obtained from six bioinformatics websites. Subsequently, q-PCR was employed to test all six genes, revealing two genes with significant changes, namely Fzd6 and Gcnt2. We then predicted the binding sites of these genes and validated their existence through luciferase assays. Moreover, we examined the expression of these two potential targets in human KOA samples using a human database and found them to be expressed specifically in the samples. These results suggest that Fzd6 and Gcnt2 are potential target genes for KOA. However, we didn’t do western blot assay to verify the results. Based on your suggestions, we have further discussed the limitations of our study in this regard and proposed future research strategies.

(5) To verify the binding site on 3'UTR of two potential targets, the authors designed a mouse sequence for luciferase assay, but not sure if it is the same when using a human sequence.

Thank for your great advice. We carried out the comparative analysis of sequence conservatism between human and mouse, and find the binding site on 3'UTR matches to human sequence very well. The sequence conservation between hsa_miR-199b-5p and mmu_miR-199b-5p was as high as 95.65%. We added the methods and results in the revised manuscript. (page9, line181-184; page17, line361-365) (supplemental fig6).

In detail: Firstly, the sequence information of mmu_miRNA-199b-5p was used to locate the human homologous sequence in the UCSC database. The homologous sequence was found to be located in the human genome at chr9:128244721-128244830 (supplemental fig6 A). Based on this positional information and the source gene, a further comparison was conducted in miRbase to identify the nearest miRNA at the position of the human genome. It was discovered that hsa_miR-199b-5p is positionally conserved and located at chr9:128244721-128244830 (supplemental fig6 B). The sequence of hsa_miR-199b-5p was obtained from the miRbase database (supplemental fig6 C), and a comparative analysis was performed between the sequences of humans and mouse (supplemental fig6 D). Besides being positionally conserved, the sequence conservation between hsa_miR-199b-5p and mmu_miR-199b-5p was as high as 95.65%, indicating a good sequence conservation.

**Author response image 2. sa3fig2:** Comparative analysis of sequence conservatism between human and mouse. (A) The position of its human homologous sequence in the UCSC database. (B) Position of the closest miRNA in miRbase. (C) Sequences of hsa_miR-199b-5p and mmu_miR-199b-5p. (D) Sequence conservation of miR-199b-5p.

**Reviewer #2 (Public Review):**
Summary:The authors identified miR-199b-5p as a potential OA target gene using serum exosomal small RNA-seq from human healthy and OA patients. Their RNA-seq results were further compared with publicly available datasets to validate their finding of miR-199b-5p. In vitro chondrocyte culture with miR-199b-5p mimic/inhibitor and in vivo animal models were used to evaluate the function of miR-199b-5p in OA. The possible genes that were potentially regulated by miR-199b-5p were also predicted (i.e., Fzd6 and Gcnt2) and then validated by using Luciferase assays.

We greatly appreciate Reviewer #2 constructive comments.

Strengths:(1) Strong in vivo animal models including pain tests.(2) Validates the binding of miR-199b-5p with Fzd6 and binding of miR-199b-5p with Gcnt2.

Thanks for positive comment.

Weaknesses:(1) The authors may overinterpret their results. The current work shows the possible bindings between miR-199b-5p and Fzd6 as well as bindings between miR-199b-5p and Gcnt2. However, whether miR-199b-5p truly functions through Fzd6 and/or Gcnt2 requires genetic knockdown of Fzd6 and Gcnt2 in the presence of miR-199b-5p.

In this study, we employed a comprehensive approach by integrating data from six bioinformatics databases to identify potential target genes for miR-199b-5p. Subsequent qPCR analysis revealed significant changes in two genes, Fzd6 and Gcnt2. We then utilized luciferase assays to validate the predicted binding sites and confirmed the interaction between miR-199b-5p and these genes. Additionally, we examined the expression profiles of these potential target genes in human KOA samples using a human database, which unveiled distinct expression patterns.

While our findings suggest that Fzd6 and Gcnt2 may serve as potential target genes for miR-199b-5p, we acknowledge the necessity for further experimental validation and in-depth functional characterization. Building upon your insightful recommendations, we have thoroughly addressed the research limitations and proposed potential research strategies for future investigations in our discussion. (page11，line227-231)

(2) In vitro chondrocyte experiments were conducted in a 2D manner, which led to chondrocyte de-differentiation and thus may not represent the chondrocyte response to the treatments.

We admit that 3D culture system will be more accurate and reliable. However, according to Liu Qianqian et al researches3, the 2D culture systems were also used and work well. Besides, the second-generation primary mice chondrocytes we used in the current study did not exhibit a significant dedifferentiated morphology. So, considering the experiment condition in our lab, we chose the second-generation cultured primary mouse chondrocytes in the whole process of cell experiment. To show the reliability of the cells, we provided more pictures in the supplement fig 7(fig. S7) In the future study, we will adopt 3D culture system for experiments. Thank you for your advices and we have added this limitation in the revised manuscript. (page11，line237-240)

**Author response image 3. sa3fig3:** Primary mice chondrocytes we cultured (P1) and the secondary generation cells (P2) we used in the following experiment.

References which used 2D ：

(3) Liu Q, Zhai L, Han M, et al. SH2 Domain-Containing Phosphatase 2 Inhibition Attenuates Osteoarthritis by Maintaining Homeostasis of Cartilage Metabolism via the Docking Protein 1/Uridine Phosphorylase 1/Uridine Cascade. Arthritis & Rheumatology (Hoboken, NJ). 2022;74(3):462-474.

(3) There is a lack of description for bioinformatic analysis.

Sorry for our neglection. We have added relevant descriptions and details. (Pages 14, line299-303)

(4) There are several errors in figure labeling.

We have revised. (Fig. 3, Fig. 4, Fig. 5 and Fig. 7)

**Recommendations for the authors:**

We appreciate the reviewers' feedback as we believe it has significantly contributed to the refinement of our manuscript. We are confident that our revisions have strengthened the quality and impact of our study, and we agree that the suggestions presented by the reviewers are valuable and appropriate for publication.

**Reviewer #2 (Recommendations For The Authors):**
I would like to thank the authors for investigating the functional role of miR-199b-5p in knee OA. While this study has the potential to provide valuable knowledge to the fields of miRNAs and joint diseases, significant improvements in several areas are required.

We appreciate your constructive comments, and we have made a substantial improvement to the manuscript. We thank all the reviewers for their advice as well as their criticisms.

Major concerns:(1) According to the Authors, miR-199b-5p is identified by the results from their own miRNA-sequencing as well as comparison with other publicly available datasets (both synovium and cartilage datasets). It is unclear to me why the synovium dataset was used here as it appears that the entire manuscript was mainly focused on chondrocytes.

Thank you for your question. As we are aware, cartilage degradation is the initial pathological change in knee osteoarthritis (KOA), which subsequently leads to other pathological changes such as synovial inflammation4. These factors are interrelated, and current research on KOA encompasses cartilage, synovium, and system inflammation et al. Therefore, when we identified a large number of dysregulated miRNAs in extracellular vesicles isolated from serum, it was crucial to determine whether these dysregulated miRNAs were also altered in cartilage or synovium. To address this, we compared our findings with publicly available databases and found a higher overlap with the cartilage cell dataset, including miRNA-199b. Consequently, we decided to focus our subsequent investigations on cartilage-related research.

Reference

(4) Hunter D, Bierma-Zeinstra S. Osteoarthritis. Lancet (London, England). 2019;393(10182):1745-1759.

(2) Also, 169 of 177 differentially expressed exosome miRNAs were intersected with differentially expressed miRNAs from OA cartilage datasets. It is surprising that in the 5 selected miRNAs for further qRT-PCR validation, 3 out of 5 were not in the exosome miRNA dataset (i.e., hsa-mir-1296-5p, hsa-mir-15b-3p, and hsa-mir-338-3p; page 5, line 109 and Fig. 1B). Isn't that selecting the miRNAs that both differently expressed in exosome and cartilage datasets for validation more essential? Furthermore, from the Authors' exosome miRNA dataset, only 5 out of 15 KOA patients actually exhibited up-regulated miR-199b-5p vs. health controls. Please elaborate on how the target was determined.

In fact, our study included two additional groups: the acupuncture treatment group (4 weeks of continuous acupuncture treatment) and the waiting treatment group (no intervention, followed by acupuncture treatment after 4 weeks), in addition to the healthy control and knee osteoarthritis (OA) patient groups. After comparing these four groups, we found that 11 genes (hsa-miR-504-3p, hsa-miR-1915-3p, hsa-miR-103a-2-5p, hsa-miR-887-3p, hsa-miR-1228-5p, hsa-miR-34c-3p, hsa-miR-3168, hsa-miR-518e-3p, hsa-miR-1296-5p, hsa-miR-338-3p, and hsa-miR-199b-5p) were upregulated in KOA patients but downregulated after acupuncture treatment, with no change in the waiting treatment group. Additionally, 7 genes (hsa-miR-448, hsa-miR-514a-3p, hsa-miR-4440, hsa-let-7f-5p, hsa-let-7a-5p, hsa-let-7d-5p, and hsa-miR-15b-3p) were downregulated in KOA patients but upregulated after acupuncture treatment, with no change in the waiting treatment group. Considering the improvement in clinical symptoms of KOA patients after acupuncture treatment, we believe that these 18 genes are of significant value. Based on overall expression abundance and species specificity, we finally selected 5 genes, namely the 5 genes mentioned in this article. Regarding this result, we have already included it in the supplementary fig5(fig. S5).

**Author response image 4. sa3fig4:** Venn diagram showing differentially expressed miRNAs in the OA group compared with healthy patients and patients who recovered after acupuncture treatment.

(3) There is also a lack of description for bioinformatic analysis regarding how miRNA sequencing datasets were analyzed. What R/python packages or algorithms were used? What were the QC criteria?

We apologize for any confusion caused. We have now included a clear description of the method employed, and R was utilized for this data analysis (revised in Page14, Line301-305). To ensure consistency, we compared our findings with publicly available human serum data from the database (GSE105027) using a fold change threshold of > 1.5 and a significance level of p < 0.05. In the cartilage data (GSE175961), we observed a list of miRNAs with shared expression patterns, yet the precise differential values could not be determined.

(4) Another major concern is the chondrocyte culture method. Chondrocytes should be cultured in a 3D manner (i.e., a 3D pellet culture system or a micro mass culture method). 2D cultured chondrocytes tend to de-differentiate into MSC-like cells and thus lose their chondrocyte phenotype. This is evident from Fig. 3B and C. Cells started to spread out and only a few cells were positive for COL2A1 with a deep brown staining color. Thus, the results from the in vitro studies may not be representative of chondrocyte response to the treatments.

We admit that 3D culture system will be more accurate and reliable. However, according to Liu Qianqian et al researches3, the 2D culture systems were also used and work well. Besides, the second-generation primary mice chondrocytes we used in the current study did not exhibit a significant dedifferentiated morphology. So, considering the experiment condition in our lab, we chose the second-generation cultured primary mouse chondrocytes in the whole process of cell experiment. To show the reliability of the cells, we provided more pictures in the supplement fig 7(fig. S7) In the future study, we will adopt 3D culture system for experiments. Thank you for your advices and we have added this limitation in the revised manuscript. (page11, line237-240)

**Author response image 5. sa3fig5:** Primary mice chondrocytes we cultured (P1) and the secondary generation cells (P2) we used in the following experiment.

References which used 2D ：

(3) Liu Q, Zhai L, Han M, et al. SH2 Domain-Containing Phosphatase 2 Inhibition Attenuates Osteoarthritis by Maintaining Homeostasis of Cartilage Metabolism via the Docking Protein 1/Uridine Phosphorylase 1/Uridine Cascade. Arthritis & Rheumatology (Hoboken, NJ). 2022;74(3):462-474.

(5) Page 7, lines 148-149: "The cartilage of mice injected with the miR-199b-5p mimic was slightly degraded (p=0.02) (Fig. 4E, F)". However, there was no significance between the groups found in Fig. 4F. Also, from the histological images of Fig. 4E, it looks like mice with inhibitor injection had more cartilage damage than miR-199b-5p mimic.

We apologize for any confusion caused. Figures 4E and 4F represent the Safranin Fast Green Staining staining of the joint after the administration of miR-199b-5p inhibitor and mimic under physiological conditions. As you can see, there is minimal difference between these four images. There is no statistically significant difference. However, in Figures 5E and 5F, the MIA-induced KOA model was utilized, and noticeable differences can be observed after the administration of the inhibitor and mimic. In the revised version, we have emphasized that Figures 4E and 4F represent the results under physiological conditions, not under the MIA-induced model. (page 7, line 146-151)

(6) Page 7, lines 149-150: "Additionally, the articular surface showed insect erosion (Fig. 4G)." It is also unclear how micro-CT analysis will be able to demonstrate the erosion of cartilage. Or the authors actually indicate the trochlear groove. However, this could also be observed in the control group and the results were not quantified. It is also unclear if the cross-section images of micro-CT shown here are helpful at all without any further explanation in the manuscript.

Figure 4 G represents control, vehicle control, inhibitor, and mimic groups, while Figure 5 G represents model, model+vehicle control, model+inhibitor, and model+mimic groups. From Figure 4G, it can be observed that the simulator group showed the most obvious erosion appearance, while the inhibitor group did not exhibit this phenomenon5. From Figure 5G, it can be seen that the model group and model+mimic group exhibited the most pronounced erosion appearance, while the model+inhibitor group showed the best recovery. To highlight the pathological changes in the erosion appearance, we marked the typical locations with red arrows in the images for easy comparison and reading by the readers (Fig. 4G; Fig. 5G). We also made corresponding textual modifications in the original manuscript to address these findings (page 7, line 150-151; page 8, line 160-161). In addition, the 3D reconstruction of micro-CT is based on the synthesis of these cross-sectional images.

References

(5) Tao Y, Wang Z, Wang L, et al. Downregulation of miR-106b attenuates inflammatory responses and joint damage in collagen-induced arthritis. Rheumatology (Oxford, England). 2017;56(10):1804-1813.

(7) Page 17, line 309-310: "Before model establishment and at 3, 7, 10, 14, 21, and 28 days after model establishment." Please re-write this as this is not clear regarding the experimental procedure.

Thank you. We had to re-write the sentences as following：Baseline testing of behavioral pain thresholds was conducted prior to model establishment, followed by behavioral pain threshold testing on days 3, 7, 10, 14, 21, and 28 after model establishment. (pages15, line322-324)

(8) Fig. 5A. The M + inhibitor and Model images are not at the same plane as M + mimic and M + RNAnc images.

Thank you. We have modified.

(9) Fig. 5B. There are two lines both with circle markers (Control and M+inhibitor). Please correct.

We have corrected.

(10) Fig. 5F. Missing * sign.

We added *sign.

(11) Please elaborate how the potential binding sites between miR-199b-5p and Gcnt2 and between miR-199b-5p and Fzd6.

We apologize for any lack of clarity in the original text. In fact, we utilized targets to predict potential binding sites. Specifically, for the mouse species, we predicted that the 3'UTR of Fzd6 binds with miR-199b-5p at positions 2483-2490, 3244-3251, 3303-3309, and 3854-3860, while the 3'UTR of Gcnt2 binds with miR-199b-5p at positions 2755-2762 and 4144-4151. In the revised version, we provide a detailed description of the methodology used for predicting these sites and offer an elaborate explanation of the results. (pages16, line352)

Additionally, to demonstrate consistency with human binding sites, we not only predicted the binding sites of human miR with these two target genes but also found a high conservation of up to 95.65% between the human and mouse sequences of miR-199b-5p. We have included this information in the supplementary materials (Fig. S6).In Fig. 6E-F, we presented the potential binding sites between miR-199b-5p and Gcnt2, as well as between miR-199b-5p and Fzd6. In addition, we provide the predicted binding of human sequence to illustrate the binding sites. Furthermore, the predicted binding of human miR-199b-5p with fzd6 and gcnt2 showed a high degree of consistency. (The fluorescent labeling in the following text indicates the potential predicted binding sites.) (Supplement file 8)

hsa-miR-199b-5p MIMAT0000263

CCCAGUGUUUAGACUAUCUGUUC

NCBI Gene ID 8323 GenBank Accession NM_001164615

Gene Symbol FZD6 3' UTR Length 1368

Gene Description frizzled class receptor 6

3' UTR Sequence: agaacattttctctcgttactcagaagcaaatttgtgttacactggaagtgacctatgcactgttttgtaagaatcactgttacattcttcttttgcacttaaagttgcattgcctactgttatactggaaaaaatagagttcaagaataatatgactcatttcacacaaaggttaatgacaacaatatacctgaaaacagaaatgtgcaggttaataatatttttttaatagtgtgggaggacagagttagaggaatcttccttttctatttatgaagattctactcttggtaagagtattttaagatgtactatgctattttacttttttgatataaaatcaagatatttctttgctgaagtatttaaatcttatccttgtatctttttatacatatttgaaaataagcttatatgtatttgaacttttttgaaatcctattcaagtatttttatcatgctattgtgatattttagcactttggtagcttttacactgaatttctaagaaaattgtaaaatagtcttcttttatactgtaaaaaaagatataccaaaaagtcttataataggaatttaactttaaaaacccacttattgataccttaccatctaaaatgtgtgatttttatagtctcgttttaggaatttcacagatctaaattatgtaactgaaataaggtgcttactcaaagagtgtccactattgattgtattatgctgctcactgatccttctgcatatttaaaataaaatgtcctaaagggttagtagacaaaatgttagtcttttgtatattaggccaagtgcaattgacttcccttttttaatgtttcatgaccacccattgattgtattataaccacttacagttgcttatattttttgttttaacttttgttttttaacatttagaatattacattttgtattatacagtacctttctcagacattttgtagaattcatttcggcagctcactaggattttgctgaacattaaaaagtgtgatagcgatattagtgccaatcaaatggaaaaaaggtagttttaataaacaagacacaacgtttttatacaacatactttaaaatattaaggagttttcttaattttgtttcctattaagtattattctttgggcaagattttctgatgcttttgattttctctcaatttagcatttgcttttggtttttttctctatttagcattctgttaaggcacaaaaactatgtactgtatgggaaatgttgtaaatattaccttttccacattttaaacagacaactttgaatacaaaaactttgttttgtgtgatcttttcattaataaaattatctttgtataagaaaaaaaaaaaaaa

hsa-miR-199b-5p MIMAT0000263

CCCAGUGUUUAGACUAUCUGUUC

NCBI Gene ID 2651 GenBank Accession NM_001491

Gene Symbol GCNT2 3' UTR Length 2780

Gene Description glucosaminyl (N-acetyl) transferase 2 (I blood group)

3' UTR Sequence: gctattcatgagctactcatgactgaagggaaactgcagctgggaagaggagcctgtttttgtgagagacttttgccttcgtaatgttaaccgtttcaggaccacgtttatagcttcaggacctggctacgtaattatacttaaaatatccactggacactgtgaaatacactaacaggatggctgggtagagcaatctgggcactttggccaattttagtcttgctgtttcttgatgctcacctctatattagtttattgttaggatcaatgataaatttaaatgacctcagatctttgcaccagatactcatcatatacaaatgttttagtaaaaaagagaattgtagataatactgtctaggaaaataagaattaggtttctttgaagaaggaatcttttataacaccttaacagtcaccactgtgctcaaccagacagatagtgaaacagctttctgggtaattcaccaatttcctttaaaacataagctacctgaatggagaatacatcttgtttctgagtttcaacactagcatttttggcttactcatggacaaagttctgtatatagtataaagtcattaacaagaaacaggatatgctttaagacagaattcactgtctgttgcttcagtaaaaggacctcggggaataaaacatttctctcttatatgccagaatgtaggctggtccctatgtcatgtcttccattaagaacactaaaaagtccttgcaagaatggagatatgcattcaagagaggtgctatcacatagatctagtctgaagtctggaacactttcctcttctatgacccctctctccccagtattatcttacttgcaaaatggagaccaaattctatcctgtgaggcttttaattgcaccatagtatgctctgagtagctttacactgcctggtactgatagtagtggctcgatttttaagagccttcaattgtagatgaacatctctgttatttatccctcattcatccatccgttcattcattcagccttcaatcaacatctcttgagtgtctattatgtacaggacatgtactgagacaaaaaggaaacataagagctttttcactctaaaaatcttggcaataatgtcaacaccagaaagcctcctctggagaatcttacagagtgattgtagtttaatacaggaacacacagggctgtgtagcatgataccaggcccaggagatcagtaattacaaattaagggttaaatcagagattattcaacagagagggagaaaggaggagacagagggaggacctgttgtgttccagccattctggtattcctttatgtatctaatttcattcaaacctcacaacagtcttgtgaggcccttatataattactcccattttgcagatgaagtaactgaggcttagaaaggttaatagcaccggggaacaatttctctgggtgagaattgggactctgttgctggtcttctcagttcatttcctgaggtggatttactgagagaaggtgaaataaagccatatttagtataccagagaaggtagattttaagaatggtctcagtgttaatactgagaaaaagtcctgtcagttcagaaaaaatgtgaagtctactttagtattcctgtaatactaaaccgttgagtttctaaatatttatttattctaacaaaaagcaattactacaaatggatgacacatttaatgaacacaattttattttttttctgtaactgtgcttgttgaatgtcaatcatatttaaagggaatgactttgaagtaaaaccttttttcttgctactgaaaaaaatggagttgttttgggtggtaaagtgttaaggaatagggacagctggtcacacaaggaactcttgaaggccacatgtgaaaacctgtcacttgcacagaggccagtcccactaaggtgaccagagtgggctccaagcacaaactgccattggctatagatgggactgtgtccccccaaaattcatgtgttggagccttaaccctcaatgtgatggtatttgagatggggcctttggtaagggaagtttagatgaggtcacgagggtaggaccctcatgatgggatgagtccccttacaagacctctggcttgggccgggcgtggtggctcacacctgtaatcccaacactttgggaggccaaggcaggtagatcacttgatgccaggagttccagaccaggctggccgacatggtgaaaccccatctctactaaaaaatataaaaattagccgggctttgtggcatgtgcctgtaatcccagctatttggcaggctgaggcatgagaatcgcttgaacccaggaggtggaggttacagtgagctgagagtgccccactgcactccagcctgggtgacagagcgagactttgtcccaaaacaaaataggtgaggggatagcgaatgcactcagggtcagcagtggagtttaaaaattgtctcttttcaacttatttaaatgacagcacctgagaagaggaaccgttttacactggatgtttctcatgtagaacaagaaatctttctggaattgatgtttacatgtctgttgttggtcatctctcctgtgtcttaaatactttaatgttggaagagcatagtgtttgggctagtgggtttctgacagcccatgggaatgccctgaaactactgtatctgatgtttgttttcgatgaggttccatgttttgttttcttgggaataaattaatatattgttttccaaaaaaaaaaaaaaaaaaaa

(12) Page 10-11, Line 222-223: "Our findings indicate that miR-199b-5p plays a crucial role in KOA by targeting Fzd6 and Gcnt2". This is an overstatement. The current work shows the possible bindings of miR-199b-5p and Fzd6 as well as bindings of miR-199b-5p and Gcnnt2. Whether miR-199b-5p truly functions through Fzd6 and/or Gcnt2 requires genetic knockdown of Fzd6 and Gcnt2 in the presence of miR-199b-5p. Thus, please tune down this statement and the title of the manuscript.

We agree your opinion of our conclusion. Therefore, we delete the overstatement sentences and tune down the conclusion of the manuscript. (the title; page 8,179; page11, line227-228)

(13) The Schematic figure (the last figure). Please remove osteophyte as this was not quantified in the study.

We modified the schematic figure accordingly.

Minor concerns:(1) Most figures were distorted.

We provide a new version of the figure to avoid distortions.

(2) Providing GO term numbers in Fig. 1C is not very helpful. Maybe show the GO term and corresponding numbers in the manuscript (Page 4, lines 79 - 82).

Thank you for your advice. We added the corresponding notes of the GO term numbers in the manuscript to explain each biological concept of it. (Page 4, line 77-89；Page 22，line 515-532)

(3) What were M-0.5 and M-1 in Fig. 2D? Different MIA concentrations?

Yes, these are different MIA concentrations, which we illustrate in the legend. (Page 23, line 535-536)

(4) Please follow the nomenclature of the gene symbol. For example, Fig. 3E-P should be mouse genes (?).

We modified the relevant gene symbol.

(5) Page 3, line 59. Not all chondrocytes are pathogenic cells in OA.

We are sorry for the mistake, now it has been modified. (Page 3, line 59)

(6) Typo. Page 3, line 55.

We changed the Typo.

(7) Page 4, line 78. These are differentially expressed miRNAs, not genes.

We have revised the unsuitable expression. (Page4, line75-76)

I wish the authors all the best with their continued work in this area.

Thank you for your wishes.